# Investigation on Ultrasonic Cavitation Erosion Behaviors of Al and Al-5Ti Alloys in the Distilled Water

**Jingtao Zhao** [1,2]**, Zongming Jiang** [1,2]**, Jingwen Zhu** [1,2]**, Junjia Zhang** [1,2] **and Yinglong Li** [1,2,]*

[1]  School of Materials Science and Engineering, Northeastern University, Shenyang 110004, China; 1610222@stu.neu.edu.cn (J.Z.); 1970332@stu.neu.edu.cn (Z.J.); 1800679@stu.neu.edu.cn (J.Z.); zhangjj@mail.neu.edu.cn (J.Z.)

[2]  Key Laboratory of Lightweight Structural Materials, Northeastern University, Shenyang 110819, China

*  Correspondence: liyl@smm.neu.edu.cn; Tel.: +86-024-8369-1570

**Abstract:** Al and Al-5Ti alloys were manufactured by an ultrasonic casting method with a new device, and their ultrasonic cavitation erosion behaviors of Al and Al-5Ti alloys in the distilled water were clarified. The damage mechanism was analyzed by macro photograph, scanning electronic micrograph and three-dimensional morphology, and the results demonstrate that Al-5Ti alloys have better cavitation erosion resistance than Al in terms of the mass loss and the surface damage. The deformation mechanism of Al and Al-5Ti alloys under cavitation erosion is mainly dislocation slip, and the $Al_3Ti$ phase enhances the cavitation erosion resistance of Al-5Ti alloys. In addition, the maximum depth of cavitation pits in the Al-5Ti sample is less than that in the Al sample for 31.3%.

**Keywords:** Al alloys; cavitation erosion; cavitation pit; micromorphology

---

## 1. Introduction

As a common natural phenomenon in the hydrodynamic environment, cavitation erosion has attracted considerable attention because of the surface damage it causes, the material erosion, the inefficiency and also the great vibration and noise [1–4]. Cavitation erosion is typically caused by the formation, and subsequent collapse, of vapor bubbles in a vibrating liquid or high-speed flow liquid [5]. It affects the service life of many equipment, such as water pumps, hydraulic machines and rudders [6–8]. When the ultrasonic cavitation occurs in the liquid, the alternating changes of positive and negative pressure would make the cavitation bubble grow and collapse, resulting in a local transient high temperature [5,9]. The severe collapse of a cavitation bubble would release huge energy and cause a micro-jet with a speed of 110 m/s, which leads to a collision density as high as 1.5 kg/cm$^2$ [10,11]. Blake [12,13] indicated that the interaction among cavitation will obviously change the shape and direction of jet during the collapse. The impact force on the materials surface is formed by cavitation bubble collapse, and it would lead to the phase transition and plastic deformation of the material [14–17]. Cavitation erosion has a detrimental effect on the material performance, while having a positive effect on the surface hardness [18,19], and yet there is no suitable mechanism that can perfectly explain this phenomenon at present [20,21].

Researching the cavitation erosion mechanism is the fundamental way to solve this problem. The available research of the cavitation phenomenon is mainly focused on the dynamics process of cavitation bubble growth and collapse, as well as the cavitation erosion resistance [22–29]. The application of coatings and the development of new materials are the main methods to improve the cavitation erosion resistance, and the surface morphology and lattice structure of materials are proven to have important effects on the cavitation erosion resistance [30–33]. The material composition

is an important factor that determines the change of structure morphology in the process of cavitation erosion, and it can be inferred that the materials with different composition would have a different cavitation erosion mechanism [34]. Szala et al. [33] successfully prepared $Al/Al_2O_3$ and $Cu/Al_2O_3$ composites by the low-pressure cold spray (LPCS) technique, and found that surface morphology plays an essential role in cavitation erosion resistance. Man et al. [35] studied the cavitation corrosion behavior of laser surface alloying of $Si_3N_4$ on AA6061, and found that the cavitation corrosion resistance of 100% $Si_3N_4$ alloys was three times higher than that of AA6061 alloys. Zhang et al. [36] coating NiCrSiB on Monel 400 alloys by laser cladding technology enhanced the hardness and cavitation corrosion resistance of Monel 400 alloys. Kim et al. [37,38] studied the cavitation corrosion resistance of alloys with different material compositions, such as Al-Cu, Al-Mg and Al- Si-Mg alloys.

Al and Al alloys are widely used in ships, machinery, hydraulics and other industries because of their high specific strength, outstanding ductility and excellent corrosion resistance [37]. Cavitation erosion behaviors often take place in the automotive components, such as cylinders, pistons, valves and combustion chambers, and these are mainly produced by casting or forging methods. Compared to the forging process, the casting process is superior due to its high productivity and low cost [39]. For casting alloys, in general, it was found that the cavitation erosion resistance of Al alloys is strongly affected by several microstructural properties, such as grain size, number of interfaces between different phases, presence and morphology of secondary phases [40]. It should also be mentioned that the defects such as inclusions and porosities often occur during the casting process, which could represent preferential site for erosion damage [40–42]. In order to solve these problems, an increasing number of experts in this field have focused on the advancement of casting processes, such as squeeze casting, thixocasting and rheocasting, which could guarantee high properties and reduce the porosity that have been introduced in the casting processes [43,44]. Moreover, Pola et al. [45] found that the cavitation erosion resistance of $AlSi_7$ alloys was improved by the ultrasonic treatment. Chen et al. [46,47] reported that ultrasonic treatment could reduce the porosity in the alloys, resulting in a better microstructure and improved mechanical properties.

At present, casting alloys are widely used as components which can be subjected to cavitation erosion, but the cavitation erosion resistance of casting alloys is rarely reported [39]. In this paper, the Al and Al-5Ti alloys were prepared by an ultrasonic casting method with a new device. The microstructure evolution of Al and Al-5Ti alloys in ultrasonic cavitation erosion is studied, and the effects of erosion time on the change of surface characteristics were systematically analyzed to offer a better understanding for the damage mechanism of cavitation erosion.

## 2. Materials and Methods

### 2.1. Materials

The commercial pure Al and high-purity Al-5Ti alloys were used as the cavitation materials. The pure Al and high-purity Al-5Ti alloys were prepared by a new device. The specific preparation process of high-purity Al-5Ti alloys was as follows: the pure Al ingot (9.5 kg) was added into the ceramic crucible, the melt temperature was set as 700 °C and the melt holding time was 10 min. Ti powder (0.5 kg) was gradually added into the Al melt from the feed inlet at a rate of 100 g/min, after the ultrasonic transmitter was turned on. An ultrasonic field with a frequency of 20 kHz was applied and turned off after the Ti powder was completely added for 5 min. Finally, the melt was poured out from the bottom liquid outlet, and 10 kg high purity Al-5Ti alloys were obtained after condensation. The preparation process of pure Al is the same as that of Al-5Ti alloys. Subsequently, the experimental samples ($\varphi$10 mm × 5 mm) were cut from the obtained ingot by the wire cutting machine. Table 1 shows the chemical composition of the commercial pure Al and Al-5Ti alloys. Since the rough surface of materials is very prone to cavitation pits, the samples were polished to a roughness of Ra = 0.02 micrometers before ultrasonic cavitation experiments. The experimental temperature and solute concentration also have certain influences on the cavitation behavior. Therefore, the distilled

water was used as the liquid medium in this study, and temperature control was maintained to eliminate the influence of liquid medium properties on cavitation erosion. The weight of samples before and after ultrasonic cavitation erosion were measured three times with a standard analytical balance for an average value [14].

**Table 1.** Chemical composition of the commercial pure Al and Al-5Ti alloys (wt.%).

| Aluminum | Fe | Si | Zn | Ti | Ga, S, C, Mg, Mn | Al |
|----------|--------|--------|--------|--------|------------------|---------|
| Al | 0.1153 | 0.1140 | 0.0225 | - | 0.1028 | Balance |
| Al-5Ti | 0.1924 | 0.2458 | 0.0353 | 5.0165 | 0.1871 | Balance |

*2.2. Cavitation Erosion Methods*

According to the original vibration stand, the cavitation erosion experiments were performed in ultrasonic vibratory equipment, as shown in Figure 1 [39,48]. The ultrasonic vibratory frequency was 20 kHz, the peak amplitude was 60 μm, and the distance between the samples surface and ultrasonic probe was 0.5 mm. The air compressor was used to cool the ultrasonic transducer, while the temperature of water was controlled at 25 °C by a temperature control system. The mass loss of specimens with cavitation erosion time increasing from 5 min to 200 min was measured, and the mass loss rate $M_{LR}$ was calculated in the following equation [2,49]:

$$M_{LR} = \frac{\Delta M}{t} \tag{1}$$

where $\Delta M$ is the mass loss (mg) and it represents the cavitation erosion time (min).

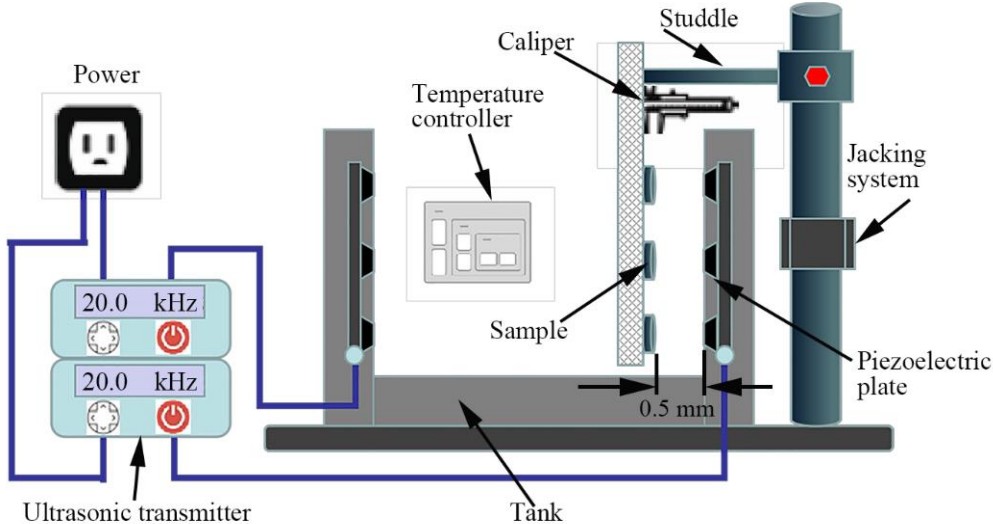

**Figure 1.** Schematic of the ultrasonic cavitation erosion device.

The advantages of the cavitation device shown in Figure 1 are that the number of test samples are increased and the complex test cycles are shortened. However, there is also an unfavorable factor, which is the layout of the station used carries the risk of deviations from the values of cavitation loads specific to ASTM G32 standard, with the risk of variation of this load during the experiment. The sources of this variability are the migration of bubbles to the liquid surface and the vibration of the sample carrier plate. Fixing a long arm of the carrier on a bracket with a relatively long arm can cause a vibration of the order amplitude, which slightly changes the distance between the sample and the ultrasonic probe.

## 3. Results and Discussion

### 3.1. Mass Loss

The mass loss of the Al and Al-5Ti samples in the cavitation erosion process is shown in Figure 2a, as well as the mass loss rate. It is obvious that the cumulative mass loss of Al and Al-5Ti is increased with cavitation erosion time (Figure 2a). The cumulative mass loss of Al is consistently higher than that of Al-5Ti, and it reaches 12.24 mg for Al and 6.11mg for Al-5Ti after cavitation erosion for 200 min. Similar to cumulative mass loss, the cumulative mass loss rate of Al is also greater than that of Al-5Ti, as shown in Figure 2b. The cavitation erosion process can be divided into four stages; namely, the incubation period, acceleration period, stabilization period and attenuation period. The incubation period of Al is quite short, and cavitation erosion accelerated after 10 min. In contrast, the cumulative mass loss rate of Al-5Ti increased slowly within 20 min, and then sharply with the extension of cavitation erosion time. The stabilization period of Al and Al-5Ti occurred at 90 min and 60 min, respectively, and the stabilization period of Al was shorter. After cavitation erosion for 200 min, the cumulative mass loss rate of Al and Al-5Ti are 0.061 mg/min and 0.031 mg/min, respectively. Apparently, ultrasonic cavitation erosion damage of Al in distilled water is more serious.

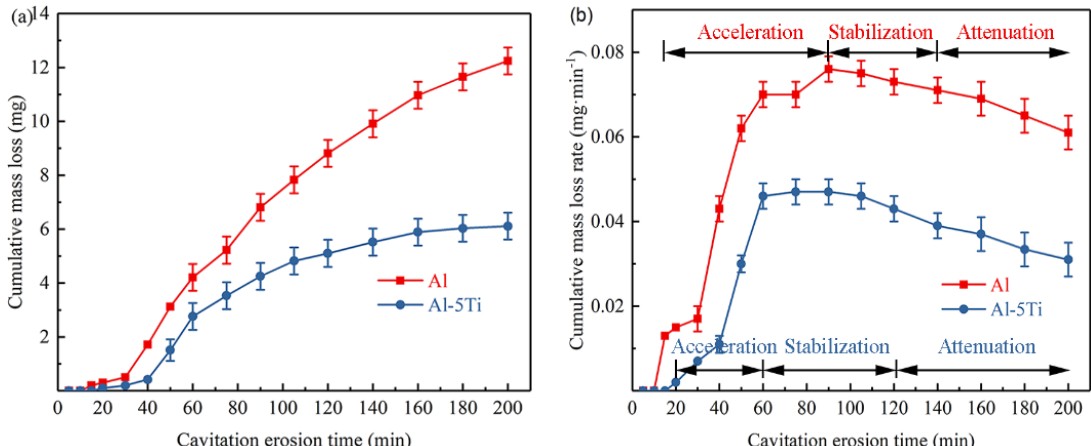

**Figure 2.** The cumulative mass loss graphs (**a**) and the cumulative mass loss rate graphs (**b**) in distilled water.

### 3.2. Cavitation Erosion Morphology

Figure 3 shows the macroscopic morphologies of Al and Al-5Ti samples after different cavitation erosion times. When cavitation erosion time was 30 min, slight cavitation pits appeared on the edge of the Al sample (Figure 3a), and they became more serious, advancing to the center with the cavitation erosion time increasing (Figure 3b). The Al-5Ti sample presents a better corrosion resistance, and few pits can be observed after 30 min of cavitation erosion (Figure 3f), while the cracks began to appear after cavitation erosion for 50 min (Figure 3g). When the cavitation erosion time reached 90 min, the Al-5Ti sample had severe cross cracks appearing on its surface, but its surface damage was still less than in the Al sample, as shown in Figure 3c,h. Additionally, the cavitation erosion area of each sample occurred first at the edges. Compared with the center of the sample, there are two reasons for cavitation erosion at the edge of the sample. One is that the cavitation erosion of the edges of the samples is likely to be caused by the implosion of the bubbles migrating towards the liquid surface. The other is that the bubbles generated at the edges are more likely to burst than that at the center of the samples.

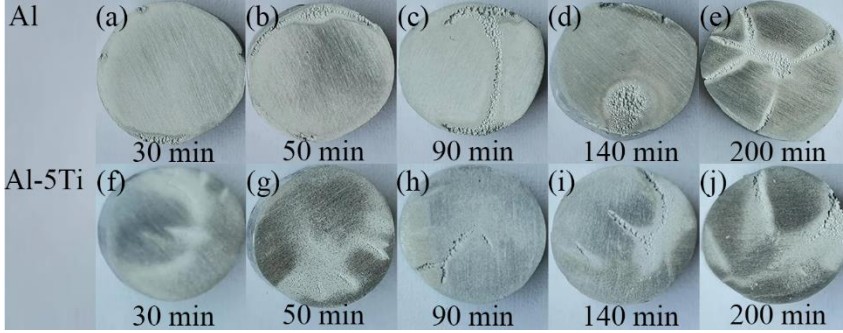

**Figure 3.** The macroscopic morphology of Al and Al-5Ti alloys after different cavitation erosion time (**a**–**e**) Al, (**f**–**j**) Al-5Ti.

The OM photos of Al and Al-5Ti samples after cavitation erosion for 50 min and 90 min are presented in Figure 4, and its corresponding surface profile is on the right. It can be seen that the surface of Al is more serious after the same cavitation erosion time, and the surface damage area is more than that of Al-5Ti. The ratio of the cavitation area of Al and Al-5Ti in the same area after 90 min of cavitation erosion measured by the metallographic method are 55.54% and 26.41%, respectively. However, this is only a part of the cracks on the sample, which cannot completely prove that the cavitation erosion resistance of Al-5Ti is better than that of Al.

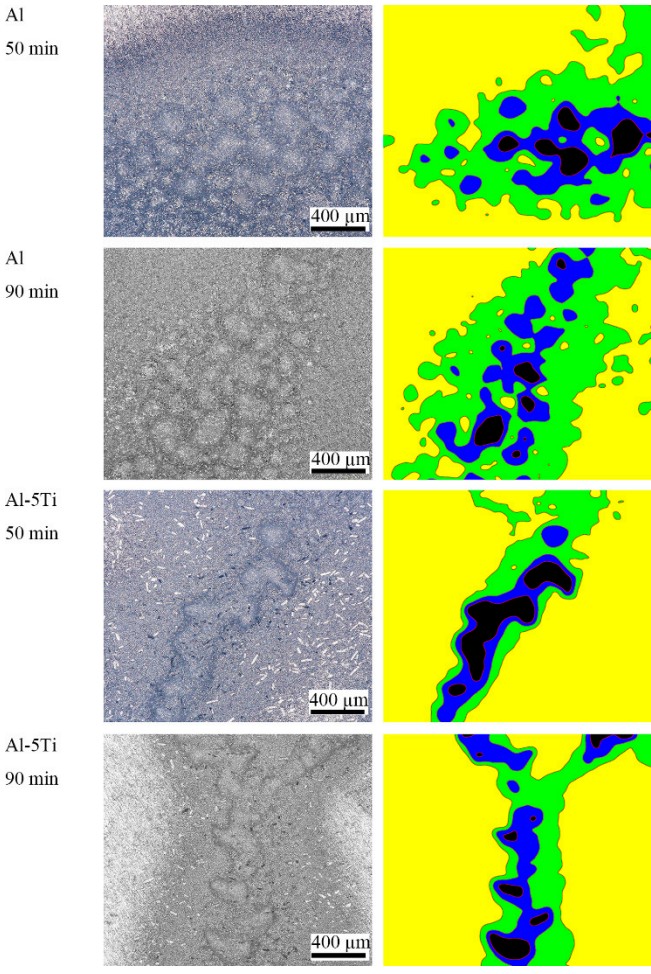

**Figure 4.** The OM photos of Al and Al-5Ti alloys after cavitation erosion for 50 min and 90 min, and its corresponding surface profile on the right.

Figure 5 shows the damaged surface of Al and Al-5Ti samples. After cavitation erosion for 30 min, the edge of the Al sample began to appear cavitation erosion, but the area of cavitation erosion is small. In contrast, darker colored areas appeared in the edges of the Al-5Ti sample, and there are no obvious cavitation pits, but the white granular material $Al_3Ti$ phase could be clearly seen. Serious cavitation pit groups appeared in both Al and Al-5Ti samples after cavitation erosion for 90 min, and the difference is that the surface damage of Al-5Ti is relatively slight. Figure 5c shows the surface damage in the form of the round cavitation pits group in Figure 3d after cavitation erosion for 140 min, and it can be seen that the cavitation pits group was formed by connecting multiple cavitation pits. After cavitation erosion for 140 min, cavitation pit cracks on the surface of the Al-5Ti sample were more serious than those after cavitation erosion for 90 min. The surface of the Al and Al-5Ti samples was seriously damaged after cavitation erosion for 200 min.

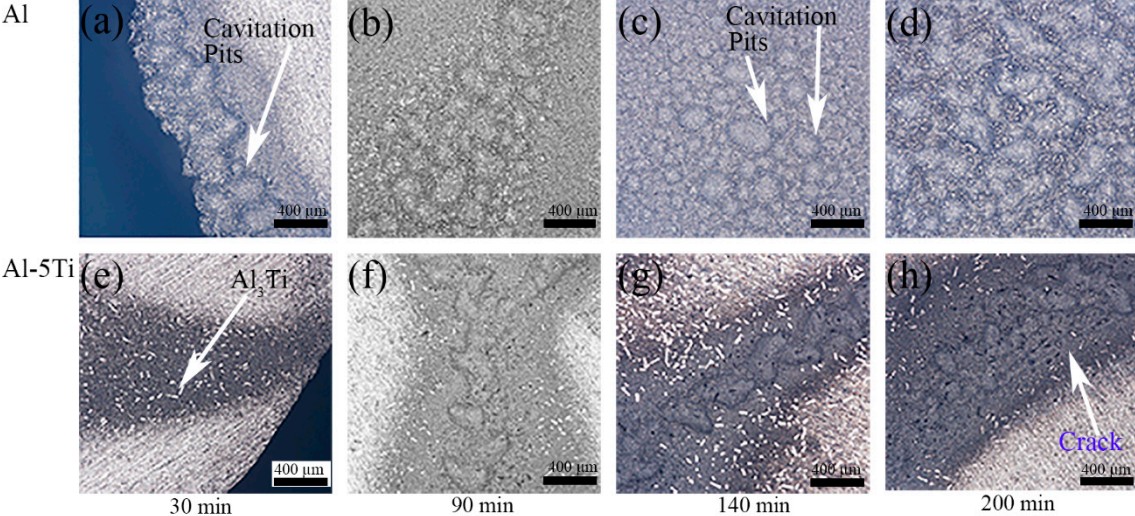

**Figure 5.** Damaged surface of Al and Al-5Ti alloys after different cavitation erosion time.

In order to study the cavitation erosion resistance of Al and Al-5Ti alloys, the erosion micromorphology after different cavitation erosion times is shown in Figure 6. The cavitation pits can be observed on the surface of Al sample after cavitation erosion for 30 min, and the sample surface became very coarse, implying heavy damage of the Al sample (Figure 6a). However, there are no obvious cavitation pits on the sample surface of Al-5Ti, and $Al_3Ti$ phases are of dispersed distribution (Figure 6c). It is a remarkable fact that the cavitation pit on the Al surface is larger than that of the Al-5Ti surface, which indicates a much more serious corrosion, as shown in Figure 6b,d. Figure 5e shows the EDS analysis results of point A in the Figure 6a,f shows the EDS analysis results of point B in Figure 6c. The results show that point A is the Al matrix and point B is the $Al_3Ti$ phases.

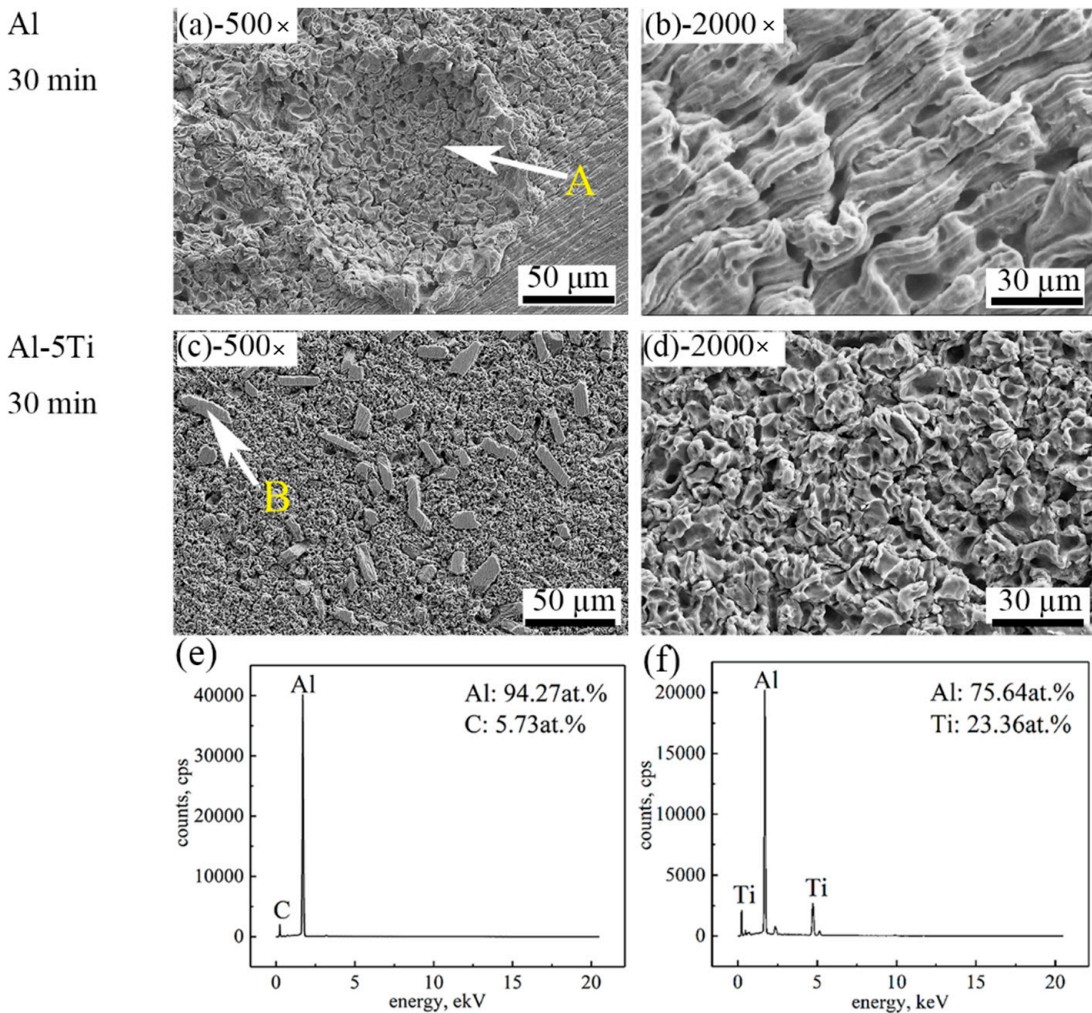

**Figure 6.** SEM images of different magnification of Al and Al-5Ti alloys after cavitation erosion for 30 min (**a**) 500× of Al, (**b**) 2000× of Al, (**c**) 500× of Al-5Ti, (**d**) 2000× of Al-5Ti, (**e**) the EDS of point A, (**f**) the EDS of point B.

Under the continuous impacts resulting from cavitation bubbles, the surface of Al and Al-5Ti samples were seriously damaged after cavitation erosion for 90 min. Figure 7a,c are SEM images of cracks in Figure 3c,h, respectively. It can be easily observed that cracks in the Al sample are longer than those in Al-5Ti. The single cavitation pit on the sample surface gradually developed into cavitation pits group after a long cavitation erosion time. The cavitation pit group then turned into a crack along the sliding direction under the impact force, which crossed longitudinally to the center of the sample. Figure 7b,d clearly show that the surface of Al and Al-5Ti samples were honeycombed with no obvious difference. Figure 8 shows the surface morphology of Al and Al-5Ti samples after cavitation erosion for 180 min. It can be obviously seen that the depth of the cavitation pit is deeper than before (Figure 8a), but the surface morphology of Al-5Ti (Figure 8c) has no significant change compared with Figure 7c. Compared with Figure 7b, it can be clearly seen that the Al grains are exposed in the cavitation pits (Figure 8b), and the Al matrix around $Al_3Ti$ is completely shed (Figure 8d) and embedded in the cavitation pit. This is a confirmation that the Al matrix in the primary site for erosion and the intermetallic particles improve the cavitation resistance of materials, as reported in previous studies by the authors [40].

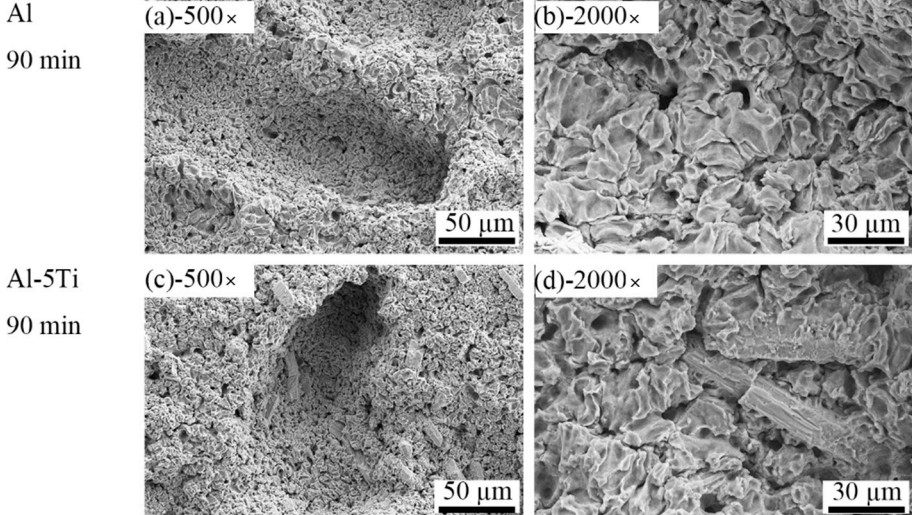

**Figure 7.** SEM images of different magnification of Al and Al-5Ti alloys after cavitation erosion for 90 min (**a**) 500× of Al, (**b**) 2000× of Al, (**c**) 500× of Al-5Ti, (**d**) 2000× of Al-5Ti.

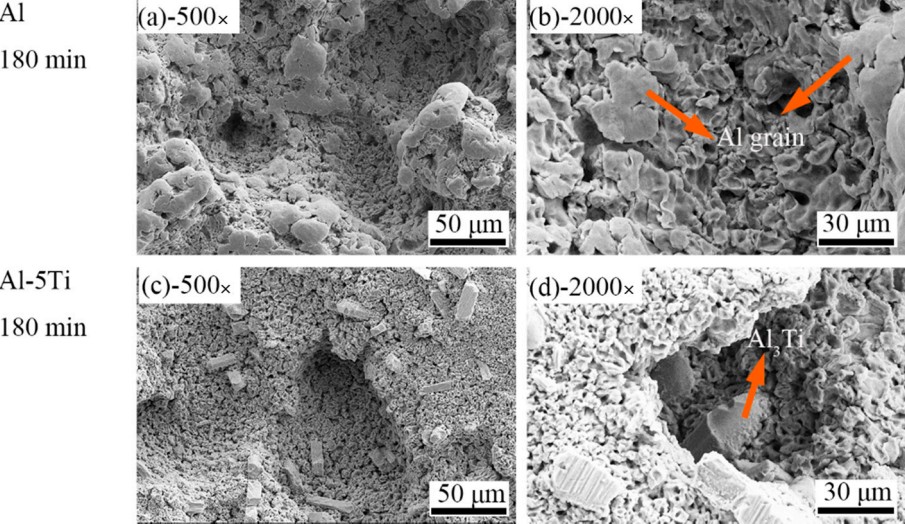

**Figure 8.** SEM images of different magnification of Al and Al-5Ti alloys after cavitation erosion for 180 min (**a**) 500× of Al, (**b**) 2000× of Al, (**c**) 500× of Al-5Ti, (**d**) 2000× of Al-5Ti.

### 3.3. Three-Dimensional Morphology

To further analyze the surface damage, the three-dimensional topographies of the Al sample after different cavitation erosion time in distilled water are shown in Figure 9. It is obvious that the cavitation pits form firstly on the edge of the sample after cavitation erosion for 5 min with a depth of 81.328 μm (Figure 9a), and then become dense after cavitation erosion 20 min (Figure 9b). When the cavitation erosion time is 40 min to 160 min (Figure 9c–g), the cavitation pits formed similar cracks in a certain direction at the center of the sample, and the area of cavitation pits region become larger. Figure 9c is the three-dimensional topography of the deepest crack on the sample, which is located in the edge area. It could indirectly prove that the crack mentioned in Figure 3 first occurs on the edge and then develops toward the center. Figure 9h shows that the sample surface exists in obvious peaks and pits after cavitation erosion for 200 min, and the final depth of the cavitation pit is 356.32 μm. In addition, the area occupied by the convex peak is very small, indicating that the material surface has severe cavitation.

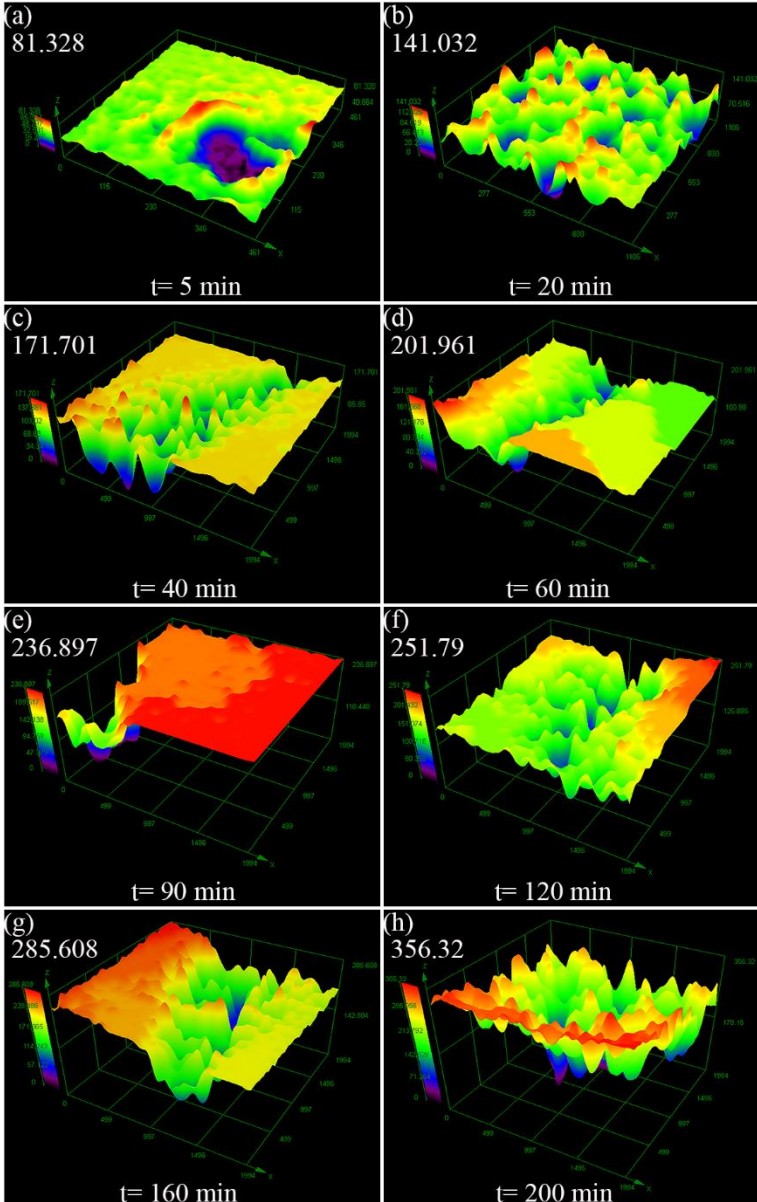

**Figure 9.** Three-dimensional morphology of Al surface after different cavitation erosion time in the distilled water (**a**) 5 min, (**b**) 20 min, (**c**) 40 min, (**d**) 60 min, (**e**) 90 min, (**f**) 120 min, (**g**) 160 min, (**h**) 200 min.

Figure 10 shows the three-dimensional morphology of the Al-5Ti surface after different cavitation erosion time in distilled water. Apparently, the sample surface remained flat with a small number of cavitation pits, as showed in (Figure 10a,b), and the depth of cavitation pits were 22.202 µm and 67.375 µm, respectively. After cavitation erosion for 40 min, no cracks appeared, but the depth of cavitation pits became significantly larger, and the sample surface began to deform. The sample surface of Al-5Ti after cavitation erosion 60 to 200 min is depicted in Figure 10d–h, and the crack is shallower than that of Al, which is similar to Figure 9. However, the cavitation pits of the Al-5Ti sample developed more slowly and appear less at the edges. In addition, the cavitation pits on the Al surface were densely distributed on the wider cracks, while the Al-5Ti surface cracks were relatively narrow with no pits. Figure 10h shows the three-dimensional morphology of Al-5Ti after cavitation erosion for 200 min, and the depth of the cavitation pit is 244.915 µm, which is 31.3% lower than Al. Figure 11 is a histogram of the maximum depth of Al and Al-5Ti cavitation pits after different cavitation erosion

times. It precisely indicates that depth of the cavitation pit is increased with the increasing cavitation erosion time, and the depth of Al-5Ti is smaller than Al.

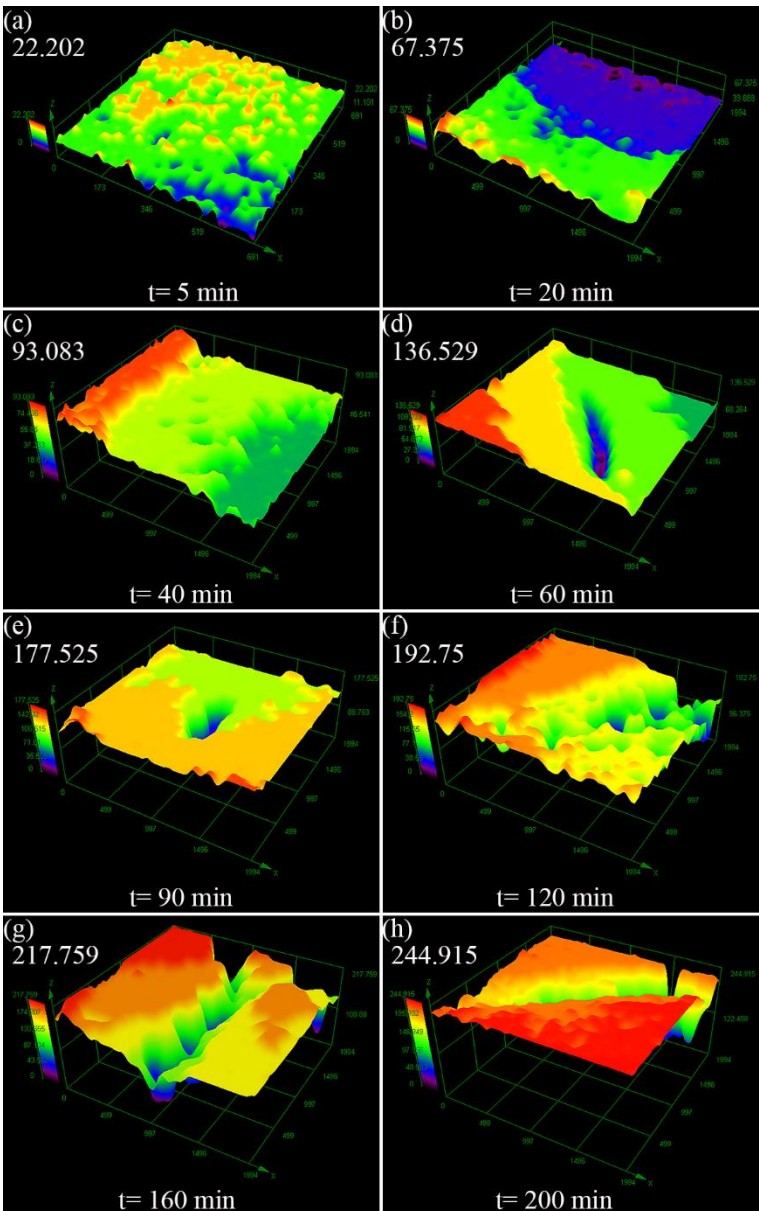

**Figure 10.** Three-dimensional morphology of Al-5Ti surface after different cavitation erosion time in the distilled water (**a**) 5 min, (**b**) 20 min, (**c**) 40 min, (**d**) 60 min, (**e**) 90 min, (**f**) 120 min, (**g**) 160 min, (**h**) 200 min.

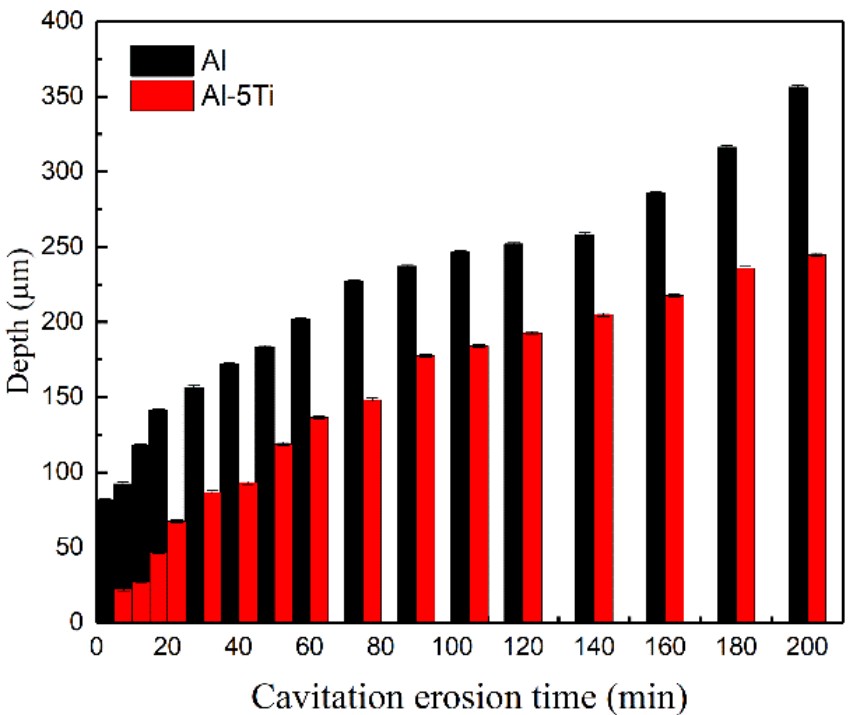

**Figure 11.** The histogram of the maximum depth of Al and Al-5Ti cavitation pits at different cavitation erosion times.

## 4. Conclusions

Al and Al-5Ti alloys were manufactured by ultrasonic casting method with a new device, and the ultrasonic cavitation erosion behavior of Al and Al-5Ti alloys in distilled water were investigated using mass loss, SEM and three-dimensional morphology. According to the research results, four conclusions were summarized as follows:

(1) During the ultrasonic cavitation experiment of Al and Al-5Ti in distilled water, the mass loss Al-5Ti samples were 6.11 mg and Al samples were 12.24 mg after cavitation erosion for 200 min. In addition, the cumulative mass loss rate of Al and Al-5Ti were 0.061 mg/min and 0.031 mg/min, respectively.

(2) The surface damage degree and cavitation erosion area of Al were more serious that of Al-5Ti in the case of the same cavitation erosion time. The ratio of the cavitation area of Al and Al-5Ti in the same area after 90 min of cavitation erosion measured by the metallographic method were 55.54% and 26.41%, respectively.

(3) Al-5Ti alloys have more resistance for cavitation erosion than Al, which is mainly because of the $Al_3Ti$ reinforcing phase and the dislocations proliferation.

(4) The three-dimensional morphology shows that the maximum depth of cavitation pits in Al-5Ti sample is less than that in the Al sample for 31.3% after the cavitation erosion.

## 5. Patents

A device for preparing high-purity Al-Ti alloys by direct reaction of Ti powder and Al melt, (patent number: ZL201610989429.7).

**Author Contributions:** Conceptualization, Y.L.; Data curation, J.Z. (Jingtao Zhao); Formal analysis, J.Z. (Jingtao Zhao) and Z.J.; Funding acquisition, Y.L.; Investigation, Z.J.; Methodology, J.Z. (Jingtao Zhao) and Y.L.; Project administration, Y.L.; Supervision, Y.L.; Validation, Y.L.; Writing—original draft, J.Z. (Jingtao Zhao); Writing—review & editing, J.Z. (Jingtao Zhao), Z.J., J.Z. (Jingwen Zhu) and J.Z. (Junjia Zhang). All authors have read and agreed to the published version of the manuscript.

**Funding:** This work was supported by the National Nature Science Foundation of China (No. 11574043).

**Conflicts of Interest:** The authors declare no conflict of interest.

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
