# Peer review of "Investigation on Ultrasonic Cavitation Erosion Behaviors of Al and Al-5Ti Alloys in the Distilled Water"

_metals, doi:10.3390/met10121631_

Round 1
Reviewer 1 Report
The paper describes the results of cavitation erosion testing of the aluminium alloy enriched with the Ti. The unique test rig was employed. The paper is worth publishing, it contains the elements of novelty. My comments on the paper:
- The authors identify the Al3Ti phase in the work for the Al-5Ti alloy (Fig. 5c, Fig. 6d, Fig 7d), but in the whole study they did not attempt to identify it, for example, by XRD or EDX. These methods should be added.
- The metallographic microstructures should be included.
- The introduction and discussion should be improved. Especially
- Fig. 8 and Fig. 9 it is difficult to identify the change of the development area (roughness) since the legends segments are on a different scale, e.g. Fig 8a max scale 22.202 Fig. Fi8b max 67.375. The red box should be reserved for the max. about 200. In such a system, it is difficult to interpret where the "0" reference level is (surface - ref level)
- Fig. 10 On the vertical axis, the unit um should be µm
- There is no discussion of the results obtained in relation to the literature data . Discussion should contain the explanation of the cavitaion erosion resitance of cast and plastic formed aluminium alloys in comparison to authors' Ti-rich alloy. The discussion should contain the comparison of the results with the literature. Following papers could be useful:
- https://doi.org/10.3390/met10070856
- https://doi.org/10.4028/www.scientific.net/SSP.227.255
- ect
- Fig. 3 is unreasonable because these are castings, e.g. for sample Al, comparing Fig3c and Fig4, the total size of the area covered by the pitting is the same and the depth of the pitting is at a similar level. The question then arises as to the number of samples and whether those presented in Fig3 are representative.
Reviewer 2 Report
The article is generally valuable especially in terms of novelty of AlTi5 alloy and technology of its production. An original test stand was also proposed to increase the number of samples tested and shorten complex test cycles.
However, it is important to answer some questions that have been imposed on me during my reading.
- Was the aim of the research only to make a semi-quantitative assessment of the cavitation resistance of the alloy or was it also interested in the stage of incubation of the damage? In this context, there is no such an important parameter in the description of sample preparation as the way of sample surface preparation and the resulting surface roughness. In the case of tests taking into account the incubation stage it is recommended to polish the samples to a roughness of Ra<0.02 micrometer .
- The development and distribution of damage is significantly different from that, which is observed during the "canonical" execution of the test ( Axially symmetrical damage zone, spreading from the center of the specimen). Is such a development of the damage a structure of influence of microstructural factors of the cast samples, or is it a result of individual features of the test stand? The features that influence the course of the destruction are indicated as follows
- The method of fixing the samples (the description is not presented) and the state of internal stresses introduced by the fixing.
- Features of the cavitation cloud (from the presented diagram it can be seen that the cloud may be asymmetrical. is it certain that the vibrations of the plate transducer are 'flat'?
- The stiffness of the sample carrier plate raises doubts - is it susceptible to vibrations? information about the vibrations of objects induced by cavitation is available in publications, e.g. on ship technology.
- All of the above factors may affect the unevenness of cavitation load - have differences in damage development been observed for upper, middle and lower samples ?
Best regards
Reviewer 3 Report
The proposed article contains several inconsistencies:
- it is not explained why the Al-5Ti alloy was chosen,
- certain terms (such as blowhols and poor castings) are not used in the scientific literature,
- the description of investigative methods is missing; “New device” does not belong to appendix A, it can be described in the text or references can be given,
- the magnifications of the microstructures in Figures 5, 6 and 7 are not appropriate,
- In general, the article is written incomprehensibly.
Round 2
Reviewer 1 Report
Dear Authors,
Thank you for your explanations.
Some of your answers and improvements (my comments 1 and 2) still need further explanations.
Comment 1: You claim the Al-5Ti phase. Please support your point of view by the literature references or XRD results, or add the metallographic etched microstructure and explain the macro- and microstructure of the castings. Please improve it.
Comment 2: I meant the metallographic microstructures of the investigated alloy. Please add it with the metallographic interpretation. Mark in the photos the important phases. Then compare it, in the discussion section, with the results of the cavitation damage.
The metallographic invesitgations should be added to the paper to confirm the presence Al-5Ti phase or it should be confirmed with usage of the scientific literature.
Reviewer 2 Report
Dear authors
Detailed reviewer's comments are presented in the attached file

Round 3
Reviewer 1 Report
Authors responses are acceptable. However, by metallographic microstructures I meant the metallographic - etched-polished surfaces which names the alloy phases. So, I think that you wrongly named "metallographic structures" while they are just a high resolution photos that magnified the damage areas. Please improve it.
You can consider adding real metallographic photos made with metallographic optical microscope, in current or in your future works.
Reviewer 2 Report
The presented version of the manuscript takes into account the reviewer's comments.
Author Response
Thank you very much.